# Assessment of the In Vitro Cytotoxicity Effects of the Leaf Methanol Extract of *Crinum zeylanicum* on Mouse Induced Pluripotent Stem Cells and Their Cardiomyocytes Derivatives

**DOI:** 10.3390/ph14121208

**Published:** 2021-11-23

**Authors:** Magloire Kanyou Ndjenda II, Elvine Pami Nguelefack-Mbuyo, Jürgen Hescheler, Télesphore Benoît Nguelefack, Filomain Nguemo

**Affiliations:** 1Research Unit of Animal Physiology and Phytopharmacology, Faculty of Science, University of Dschang, P.O. Box 67 Dschang, Cameroon; magloirendjenda@yahoo.fr (M.K.N.II); mbuyopamielvine@gmail.com (E.P.N.-M.); nguelefack@yahoo.fr (T.B.N.); 2Center for Physiology, Faculty of Medicine and University Hospital Cologne, University of Cologne, Robert-Koch-Str. 39, 50931 Cologne, Germany; j.hescheler@uni-koeln.de

**Keywords:** *Crinum zeylanicum*, pluripotent stem cells, differentiation, cardiomyocytes, cytotoxicity

## Abstract

*Crinum zeylanicum* (*C. zeylanicum*) is commonly used in African folk medicine to treat cardiovascular ailments. In the present study, we investigated the cytotoxic effect of the leaf methanol extract of *C. zeylanicum* (CZE) using mouse pluripotent stem cells (mPSCs). mPSCs and their cardiomyocytes (CMs) derivatives were exposed to CZE at different concentrations. Cell proliferation, differentiation capacity, and beating activity were assessed using xCELLigence system and microscopy for embryoid body (EB) morphology. Expression of markers associated with major cardiac cell types was examined by immunofluorescence and quantitative RT-PCR. Intracellular reactive oxygen species (ROS) levels were assessed by dichlorodihydrofluorescein diacetate staining. The results showed that the plant extract significantly reduced cell proliferation and viability in a concentration- and time-dependent manner. This was accompanied by a decrease in EB size and an increase in intracellular ROS. High concentrations of CZE decreased the expression of some important cardiac biomarkers. In addition, CZE treatment was associated with poor sarcomere structural organization of CMs and significantly decreased the amplitude and beating rate of CMs, without affecting CMs viability. These results indicate that CZE might be toxic at high concentrations in the embryonic stages of stem cells and could modulate the contracting activity of CMs.

## 1. Introduction

Biodiversity plays a significant role in human well-being and health through its impact on food and medicine production. In several countries, traditional medicine has a long history and still plays an important role in the prevention and treatment of various diseases. It is estimated that about 80% of the world’s population from developing nations still rely on plant-derived medicines for their health care [1]. In the past, there has been a growing interest in traditional medicine and its relevance to public health in developing countries. To date, several active chemicals derived from plants are used in traditional medicine [2]. The search for bioactive molecules from nature continues to play an important role in the generation of new drugs. With the appearance of the new coronavirus in 2019, known as the SARS-CoV-2, and as the world looks towards science for the development of a truly effective therapy, several countries with a long history of traditional medicine practice (such as many African countries) have increased their interest in herbal/plant medicines. Nevertheless, the therapeutic use of plants and their derivatives has been veiled by inadequate scientific research, lack of standardization, and poor documentation [3,4]. Possible side effects are serious concerns regarding traditional medicine practice [5]. The major objections include safety, efficacy, quality, and rational use of traditional medicines. Therefore, adequate tests using sophisticated tools and techniques to accommodate wider acceptance, recognition, and utilization of medicinal plants by all parties in the scientific community should be performed.

Pluripotent stem cells offer enormous potential to develop a wide variety of cell-based test systems and constitute an ideal platform that can be used to improve our understanding of human development and how degenerative diseases develop and progress. Moreover, their ability to differentiate into any cell type found in the human body makes them an ideal tool for in vitro drug screening/toxicity testing and provides the potential for disease models to perform clinical trials with specific cell types in a dish [6].

*Crinum zeylanicum* (*C. zeylanicum)* is a bulbous plant of the Amaryllidaceae family and is widely used in African traditional medicine for the treatment of various diseases such as cardiovascular diseases, rheumatic pain, ear ache, skin trouble, injuries, malaria, and epilepsy [7]. Several compounds such as alkaloids, flavonoids, fatty acids, and terpenes have been isolated from different parts of *C. zeylanicum*. The alkaloids are the major compounds isolated and the main source of cholinesterase inhibitors [8]. Interesting antihypertensive and bradycardiac effects of CZE were recently demonstrated in Wistar rat [3]. Therefore, it became important to assess the toxicity of the leaves extract of *C. zeylanicum*.

Considering the stem cells as a good in vitro test model and CZE as a source of heart modulators, the aim of the present work was, on the one hand, to conduct comprehensive investigation on its cytotoxic effect to highlight any hidden toxic activity, and on the other hand, to examine its effects on spontaneously beating activity of CMs. This work was therefore performed to gain some insight into the action of CZE after short- and long-term exposure on cell growth/differentiation and on beating parameters using mouse pluripotent stem cells (mPSCs) and their cardiac cells derivatives.

Our results demonstrate that CZE might be toxic at high concentrations in the embryonic stages of stem cells and could modulate the contracting activity of cardiomyocytes (CMs). Therefore, detailed phytochemical profiling and compound determination of the leaf extract from CZE should be carefully performed in order to exploit the constituents and their individual toxic effects.

## 2. Results

### 2.1. Cytotoxic Effects of CZE

For the cytotoxicity test, undifferentiated cells were exposed to various concentrations (0 (control), 1, 5 10, 15, 20, and 30 µm/mL) of CZE (Figure 1A) and monitored using the xCELLigence RTCA system for up to 120 h. The cell viability and proliferation were then analyzed at different time points. As shown in Figure 1B, the viability of miPSCs was significanty reduced by CZE in a time- and concentration-dependent manner as compared to the negative control (untreated cells). After 24 h, concentrations of 20 µg/mL and 30 µg/mL exhibited a significant (*p* < 0.05 and *p* < 0.01) reduction in CI compared to lower concentrations and untreated cells. The results also showed that CZE at concentrations above 1 µg/mL induced significant (*p* < 0.05 and *p* < 0.01, Figure 1C) and concentration-dependent decreases in CI 48 h and 72 h post-treatment compared to untreated cells. CZE concentrations of 10 µg/mL (low) and 30 µg/mL (high) were selected for results confirmation. As reactive oxygen species (ROS) are important cell signaling molecules that can disrupt normal physiology at elevated concentration, the effect of CZE on ROS production was assessed after 48 h of exposure. As shown in Figure 1D, CZE induced a significant increase (*p* < 0.05) in ROS production as compared to the untreated group.

### 2.2. Effect of CZE on Cardiac Differentiation of miPS Cells

Undifferentiated miPSCs were differentiated to CMs by formation of EBs through mass culture protocol in the absence (control) or presence of CZE (CZE-treated), as described in the methods section. The morphology of the EBs and the first appearance of beating EBs were observed in untreated and CZE-treated conditions between day 7 and day 9 (data not shown). As shown in Figure 2A–C, the morphology of the EBs at day 12 post-differentiation was uniform, whereas the percentage of beating EBs was significantly (*p* < 0.05) reduced under CZE treatment (Figure 2B). Concentrations of CZE up to 10 µg/mL did not affect the number of beating EBs, while concentrations of 20 µg/mL and 30 µg/mL markedly reduced the percentage of spontaneously beating EBs. We further used the concentration of 30 mg/mL to check possible change in some important cardiac genes’ expression. Previously, the effect of CZE on the morphology of miPSCs differentiation towards CMs was also determined as illustrated in Figure 2C. It can be observed that CZE at the concentration of 30 µg/mL significantly (*p* < 0.05) reduced the size of EBs compared to the control. In addition, an exponential growth of EBs during the differentiation was noticed (data not shown). The diameter of EBs at day 12 of differentiation was 322.8 ± 31 µm for the control versus 243.6 ± 12.6 µm for EBs treated with CZE (Figure 2C).

To further elucidate the above-mentioned effect of CZE on the differentiation of miPSCs to CMs, qRT-PCR analysis was performed to investigate the expression of some important groups of cardiac homeobox genes, namely, *NKx2.5, Myh6, Myl2, Nppa,* and *cTnT*. As shown in Figure 2D, the expression level of Myh6, Myl2 and cTnT was found to be significantly reduced in CZE-treated (30 µg/mL) EBs at day 12 of differentiation as compared to untreated ones, whereas the level of NKx2.5 and Nppa was not significantly (*p* > 0.05) affected by CZE.

We next determined whether CZE could affect the number of CMs during differentiation. The expression of GFP-positive cells within the EBs provided a readout for the assessment of cardiac differentiation efficiency. Thus, single cells from day 12 dissociated EBs from untreated and CZE-treated groups were examined. CZE showed a tendency towards reducing the number of CMs, but this effect was nonsignificant compared to the untreated condition. However, we observed a highly variable number of CMs in three independent experiments (data not shown). The impact of CZE on the structural organization of differentiated CMs was determined by performing immunostaining of cardiac-specific protein, cardiac α-actinin, in CMs dissociated from EBs at day 12 of differentiation. The analysis of Figure 3 revealed less expression of GFP in CZE-treated cells as compared to untreated cells. This observation was accompanied by a reduction in sarcomere striation and organization (Figure 3 inset shows a higher magnification).

### 2.3. Effect of CZE on Spontaneous Contractile Activity of miPSCs-Derived CMs

The action of CZE on miPSC-CMs integrity and function was further investigated using the xCELLigence RTCA Cardio system to evaluate its cytotoxic effects and pharmacological properties on CMs. As shown in Figure 4A, CMs were platted on 96-well E-Plate and treated with different concentrations of CZE. CZE at concentrations below 20 µg/mL did not alter cell spreading and syncytium formation throughout the experiment period, as illustrated by the CI (Figure 4B). Yet, when concentrations of 20 and 30 µg/mL were used, a slight decrease in CI was observed after 180 h continuous exposure to CZE.

It can be observed from Figure 4C that the exposure of a synchronous beating layer of iPS cell-derived CMs to CZE induced significant changes in the beating pattern in a time- and concentration-dependent manner. At 24 h exposure, CZE produced biphasic effects on the beating rate of CMs. At lower concentrations (1, 5, and 15 µg/mL), CZE induced a progressive increase in CMs beating rate that became significant at the concentration of 15 µg/mL. However, CZE significantly (*p* < 0.05) reduced the CMs beating concentrations of 20 and 30 µg/mL after 24 h as compared to the control (Figure 4D). After 48 h of treatment, the decrease in CMs beating rate elicited by CZE started at the concentration of 10 µg/mL and was concentration-dependent (*p* < 0.01). In addition, the amplitude of the beating signal was affected by the exposure of CMs to the methanol extract of CZE. After 24 h of exposure, CZE at doses of 20 and 30 µg/mL significantly (*p* < 0.01) decreased the amplitude of the beating signal (Figure 4E). As observed with the results on the CMs beating rate, CMs treated with CZE for 48 h exhibited a marked concentration-dependent decrease in the amplitude of beating that started at 10 µg/mL CZE.

## 3. Discussion

The ability of pluripotent stem cells (PSCs) to proliferate indefinitely without losing their pluripotency capacity and to give rise to different human body cell types provides a unique possibility to develop a wide variety of cell-based test systems. Therefore, they represent a suitable tool for regenerative medicine and to assess the efficacy and effectivity of any potential compounds and to test medicinal plants used as traditional treatments for numerous diseases around the world [1,9]. Considering this advantage, we investigated the effects of *Crinum zeylanicum* methanolic extracts (CZE), a bulbous plant of the family Amaryllidaceae, widely used in traditional medicine in West Cameroon [7], using pluripotent stem cells and their CM derivatives.

Previous studies reported that extracts from *Crinum* species exert a wide range of biological activities such as antihypertensive, antibacterial, antiviral, and antiallergic treatment of neurodegenerative diseases, pain treatment, and wound healing [10,11]. Amaryllidaceae is the leading family of genera holding anticholinesterase inhibitors (AChE) activity [12]. In fact, several secondary metabolites such as alkaloids, terpenoids, polyphenols, and quinones and many further compounds from natural sources have been reported as cholinesterase inhibitors [13,14]. For example, alkaloids and flavonoids have been shown to give protection against chronic diseases. CZE is rich in flavonoids and polyphenols, as we have previously shown [7]. In this study, we used various methods to analyze the effect of CZE on multivariable indicators of toxicity, assuming mechanisms of cell degradation and covering a broad range of structural and functional changes, including cell viability, ROS production, and cellular organization and beating activity. We found that (1) the methanol extract of CZE had a concentration-dependent detrimental effect on the proliferation and viability of miPSCs; (2) exposure of miPSCs to a high concentration of CZE during the differentiation stage caused a significant reduction in size of EBs; (3) CZE-treated cells generated higher levels of ROS; (4) high concentrations of CZE negatively affected the beating rate of EBs as well as the expression of important mesodermal markers and late cardiac markers such as Nkx2.5, Myh6, Myl2, Nppa, and cTnT; (5) CZE at higher concentrations caused a disruption in the sarcomere organization as well as in the normal contractile ability of the miPSC-CMs. These results provide us with some features about the effects and underlying mechanisms of CZE against cardiovascular disease and cancer, as well as an anti-inflammatory drugs, as used in traditional medicine.

The differentiation of pluripotent stem cells (PSCs) towards EBs formation into specific lineages is closely connected to the size, and it recapitulates different aspects of organogenesis and a high degree of self-organization [15,16]. The size and shape of EBs are important parameters that affect cell proliferation, lineage specification, and commitment in vitro [17]. Therefore, our results indicate, at least in part, that a high dose of CZE treatment may significantly impact the regulation of different factors, working individually or in combination, that participate in the normal growing and differentiation process of pluripotent stem cells into CMs. Moreover, the decrease in the relative expression of mesoderm and cardiac markers and the increase of ROS production observed under CZE treatment support the above-mentioned action of this extract. Several cellular mechanisms have been suggested to explain how ROS production could lead to many disorders, including cardiac arrhythmias. ROS are often released by cells in response to stress due to some drugs and are cytotoxic [18]. We also found that CZE causes not only less expression of *cTnT* but also sarcomere disorganization. Sarcomeres are the basic contractile unit of cardiac cells, whose function depends on a highly organized structure [19]. In addition, as the beating rate, beating amplitude, contractility of cardiac muscles, and ion channels are the major parameters used to examine cardiac function and compound effects, we therefore examined the effect of CZE on the beating activity of iPSC-CMs. In addition to the above-mentioned effects, we found that the beating activity of iPS cell-derived CMs was inhibited after exposure to CZE in a concentration-dependent manner. The changes in beating activity of iPSC-CMs treated with a high concentration (30 µg/mL) of CZE may indicate arrhythmias. In fact, the increase in ROS production, the sarcomere disorganization, and less expression of TnT are parameters that clearly support this observation. These results suggest that methanol extract of CZE at a concentration ≥10 µg/mL may induce partial impairment in cardiac specification processes and alteration of its function. In fact, it is clearly known that cholinesterase inhibitors are associated with higher rates of bradycardia. Thus, the observed effect of CZE on the beating activities of CMs may be due, at least in part, to its high cholinesterase inhibitors compound content. Plants produce a high diversity of secondary metabolites, depicting a mosaic mixture of compounds associated with various chemical classes [12].

## 4. Conclusions

In conclusion, this study led us to identify a not previously described critically important functional effect of methanol extract of CZE on cell proliferation, and cardiac specification and function, thus providing, at least in part, a scientific basis to validate the utilization of CZE in the treatment of various diseases. The observed in vitro actions of CZE revealed by this study may be attributed to its high content in metabolites, such as alkaloids and flavonoids [7,20]. However, these actions cannot only be attributed to a single constituentfound in CZE extract, but might be due to the synergistic interaction of individual secondary metabolites or constituents of this plant. Further investigations will be necessary to test each component separately and in combination to clarify the potential mode of action of CZE.

## 5. Materials and Methods

### 5.1. Plant Material and Extraction

The leaves of *C. zeylanicum* were collected in the west region of Cameroon identified at the National Herbarium of Cameroon in Yaoundé under the registration number 65654/HNC. The extract was prepared as described previously [7]. In summary, the fresh leaves of *C. zeylanicum* were collected, washed, and cut into pieces and crushed. One kilogram (1 kg) of the obtained paste was macerated in 3 L of methanol for 72 h and filtered by hydrophilic cotton and Whatman n 2° filter paper to remove particles. The particle-free crude extract was then evaporated at 65 °C using a rotary evaporator. The residue obtained was dried in a ventilated oven at 40 °C for 24 h to obtain *C. zeylanicum* methanol extract (CZE) as a solid mass. CZE yield was estimated at 248 g (24.8%) and stored at 4 °C until utilization. It was prepared in cell culture medium and stored at −20 °C for subsequent use. The phytochemical analysis of CZE revealed the presence of some secondary metabolites as well as high content of polyphenols, flavonoids, and alkaloids [7,10].

### 5.2. Cell Culture and Differentiation

Conventional murine iPS cell line were used as previously described [21]. This cell line expresses enhanced green fluorescent protein (EGFP) under the control of α-MHC promoter, which specifically allows the generation of highly pure CMs. To maintain their undifferentiated state as previously described [22], miPSCs were cultured on mitomycin C-inactivated MEFs in medium composed of Dulbecco’s Modified Eagle’s Medium (DMEM), supplemented with nonessential amino acids (0.1 mM), L-glutamine (2 mM), penicillin, streptomycin (50 μg/mL each), β-mercaptoethanol (0.1 mM), leukemia inhibitory factor (LIF) (500 U/mL) (ESGRO, Millipore; Schwalbach, Germany), and 15% fetal calf serum (FCS). Cells were passaged every 2–3 days and further used for differentiation or proliferation assays. If not stated otherwise, all cell culture media and supplements were supplied by Invitrogen Life Technologies (Karlsruhe, Germany).

For differentiation, transgenic miPSCs were cultured in spinner flasks as reported previously [23]. Briefly, 7 × 10^5^ cells/mL were suspended in Iscove’s modified Dulbecco medium (IMDM), 20% FCS, 100 μM β-mercaptoethanol, 1% nonessential amino acids, 100 U/mL penicillin, and 0.1 mg/mL streptomycin sulfate and placed on a horizontal shaker inside an incubator (37 °C). After 2 days, embryoid bodies (EBs) were formed and diluted at a density of about 150 EBs/mL of differentiation medium and stirred continuously in spinner flasks (CellSpin 250, IBS Integra Biosciences). At day 9, puromycin (8 µg/mL) was added for CMs purification, and medium with fresh puromycin was changed every second day. Thereafter, all surviving beating clusters were dissociated into single CMs with 0.05% Trypsin/EDTA, and cells were filtered through a 35–50-micron filter to remove remaining clumps. Pure CMs were subsequently preplated onto petri dishes and incubated at 37 °C in 5% CO_2_ for at least 24 h. The plating medium was then removed, and the cells were washed in prewarmed PBS to remove debris and dead cells. Finally, approximately 99% of purified CMs were obtained for impedance experiments [16].

### 5.3. Dynamic Monitoring of the Effect of CZE on Cell Proliferation Using the xCELLigence RTCA System

The xCELLigence RTCA Cardio Instrument was used to monitor the effect of CZE on proliferation. As previously described [24], the xCELLigence RTCA Cardio system consists of four main components (Figure 5A). The RTCA Cardio System records impedance signals and further processes and displays the data by converting impedance value into a cell index (CI) value. The CI is derived to represent cell status based on the changes in impedance signals, which result from cell attachment to the bottom of the wells. Thus, the CI value mainly reflects changes in the number of attached cells, their morphology (Figure 5B), and beating activity and is calculated as previously described [25].

To investigate the effect of CZE on cell growth and viability, undifferentiated miPSCs were cultured in nonadherent petri dishes. On day 0, miPSCs (undifferentiated cells) were plated (at a density of 40,000 cells/well) into an E-Plate 96-well previously coated with 0.1% gelatin. The effect of different concentrations of CZE on the cellular activity (adhesion, growth, morphology, size, and structure) was investigated using xCELLigence RTCA system (Figure 5C). CZE was added 48 h post plating. About 75% of the culture medium was refreshed every two days with the corresponding compound concentration maintained.

### 5.4. Effect of CZE on Cell Growth and Cardiomyocytes Differentiation

To investigate the effect of CZE on cell growth and CMs differentiation, differentiation was performed using the mass culture protocol as previously described [16]. Briefly, embryoid bodies (EBs) were first formed in nonadherent plates by culturing 1 × 10^6^ iPS cells in 14 mL Iscove’s Modified Dulbecco’s Medium (IMDM) supplemented with 20% FCS (Invitrogen) in the absence (untreated) or presence of CZE (CZE-treated) at different concentrations as stated above. Afterwards, untreated and CZE-treated EBs were diluted to a density of about 1000 EBs/12 mL in differentiation medium. Media and CZE were changed every second or third day throughout the differentiation period. EB formation was assessed by general morphology observation and analysis of EB diameters and beating activities (Figure 5A,B).

### 5.5. Assessment of CZE Effects on Beating Parameters of Cardiomyocytes

Before seeding with CMs, each well of a cardio E-plate 96 (ACEA Biosciences) was coated with 5 µg/mL fibronectin solution and incubated for at least three hours at 37 °C or overnight at 4 °C. Pure CMs (differentiated and purified as described above) at a density of 40,000 to 50,000 were seeded in 150 µL culture medium per well of E-plates Cardio 96 (ACEA Biosciences). After seeding, the prepared plate was left undisturbed in an incubator for 3 h at 37 °C to allow for an initial cell adhesion on the bottom of each well. Subsequently, the E-plate was mounted onto the RTCA Cardio Station, placed at 37 °C in a 5% CO_2_ incubator, and CMs adhesion and beating activity monitored using the RTCA Cardio Instrument (ACEA Biosciences).

To monitor the effect of CZE on beating activity of the miPS-CMs, different concentrations (1, 5, 10, 15, 20, and 30 µg/mL) were added to the corresponding wells 48 h postseeding; the time necessary for the formation of a synchronous and regular rhythmic beating cell layer. Medium was partially (up to 85%) exchanged every 2 days, and the compound was renewed. Data collection was controlled and analyzed by the RTCA Cardio Software that allows calculation of the CMs beating parameters such as frequency and amplitude.

### 5.6. Reactive Oxygen Species (ROS) Detection

The levels of intracellular reactive oxygen species were detected using cellular ROS Assay Kit (ab113851, Abcam, Cambridge, UK), following the manufacturer’s instructions. Briefly, undifferentiated cells cultured in untreated (control) and CZE-treated conditions for 72 h were exposed to 10 µM 2′,7′-dichlorodihydrofluorescein diacetate (DCFDA, also known as H2DCFDA, DCFH-DA, and DCFH) for 30 min in the dark (to reduce photo-bleaching) at room temperature. Thereafter, the cells were washed three time with PBS, and fluorescence was examined at the excitation/emission wavelengths of 485/535 nm using a microplate reader.

### 5.7. Immunocytochemical Staining Analysis

Immunostaining assays were performed according to the protocol described previously [21]. Briefly, cells were fixed for 10 min with 4% paraformaldehyde, permeabilized in 0.25% Triton X-100 (Sigma-Aldrich) and 0.5 M ammonium chloride in 0.25 M TBS (pH 7.4), and blocked with 0.8% bovine serum albumin (BSA) for 1 h at room temperature. Subsequently, samples were incubated with primary antibodies against α-actinin (1:500; Sigma-Aldrich) and cTnT (1:500; Abcam) at 4 °C overnight and detected by AlexaFluor 555- or AlexaFluor 647-conjugated secondary antibodies (1:1000, Invitrogen). Nuclei were stained with Hoechst 33258 (1:500, Sigma-Aldrich), and samples were examined using an Axiovert 200M florescence microscope (Zeiss, Göttingen, Germany).

### 5.8. Chemicals

All substances were, if not stated otherwise, obtained from Sigma-Aldrich Chemie GmbH, Germany.

### 5.9. Data Analysis

Data are expressed as mean ± standard error of the mean (SEM) for the number of EBs or wells indicated (*n*). Statistical significance was estimated by *t*-test and one way ANOVA. Results with *p* < 0.05 were considered statistically significant.

## Figures and Tables

**Figure 1 pharmaceuticals-14-01208-f001:**
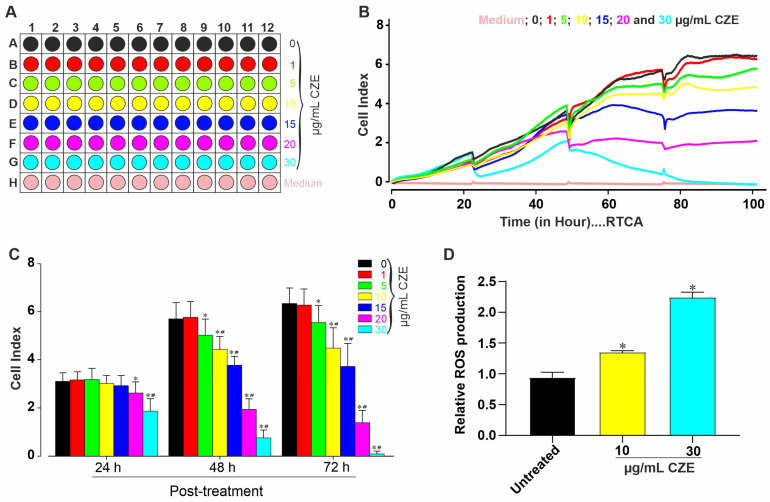
Monitoring the cytotoxicity effect and reactive oxygen species (ROS) production of the methanolic extract of *Crinum zeylanicum* (CZE) in mouse induced pluripotent stem cells (miPSCs). (**A**) Diagram showing 96-well plate design. (**B**) Representative traces showing changes in cell index, the parameter of cell viability, after 100 h of treatment with different concentrations of CZE. (**C**) Peak cell index values 24, 48, and 72 h post-treatment with 0 (negative control), 1, 5, 10, 15, 20, and 30 µg/mL CZE. (**D**) Effect of CZE on ROS production, level of ROS in untreated (control) and CZE-treated cardiomyocytes with 10 µg/mL and 30 µg/mL. Data are presented as the mean ± SEM of at least six wells for each of the three independent experiments. * *p* < 0.05 and ^#^
*p* < 0.05, significantly different compared to negative control and previous low concentration, respectively. Control wells are indicated as medium: no CMs and culture medium only.

**Figure 2 pharmaceuticals-14-01208-f002:**
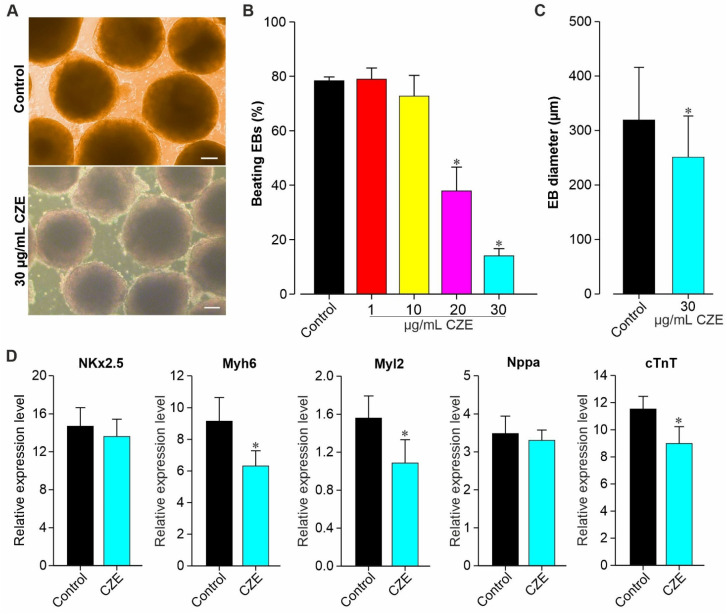
Effect of CZE on the morphology of embryoid bodies (EBs) and on cardiac marker expression genes. (**A**) Representative image of day 12 EBs cultured with 0 (control) and 30 µg/mL of CZE. (**B**) Effect of CZE concentration on the percentage of beating EBs at day 12 of differentiation. (**C**) Size (mean diameter) of control and 30 µg/mL CZE-treated EBs at day 12 post-cardiac differentiation. (**D**) Gene expression levels in CMs from control and CZE-treated EBs. Expression level of cardiac genes *NKx2.5, Myh6, Myl2, Nppa,* and *TnT* was detected using qRT-PCR. GAPDH was used as housekeeping gene and served to normalize the result. Scale bar = 50 μm. Results are reported as the mean ± SEM (*n* = 3 independent experiments). * *p* < 0.05 vs. control CMs.

**Figure 3 pharmaceuticals-14-01208-f003:**
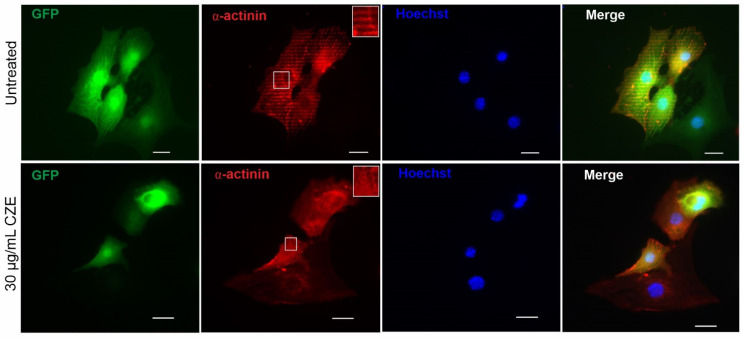
Effects of CZE on sarcomere organization of mouse induced pluripotent stem cell-derived CMs (miPSC-CMs). CMs (green) isolated from contracting areas of day 12 EBs from control (upper panel) and 30 µg/mL CZE-treated (lower panel) condition were labeled for α-actinin (**red**). Nuclei were stained with Hoechst (**blue**). Representative images shows less organized sarcomere structure in miPSC-CMs differentiated under CZE compared to those from control (untreated) condition. Merged images of GFP and α-actinin show loss of GFP in some CMs. Inset in α-actinin panel shows higher magnification of cells displaying striation organization. Scale bars = 20 µm. Abbreviation: GFP, green fluorescent protein.

**Figure 4 pharmaceuticals-14-01208-f004:**
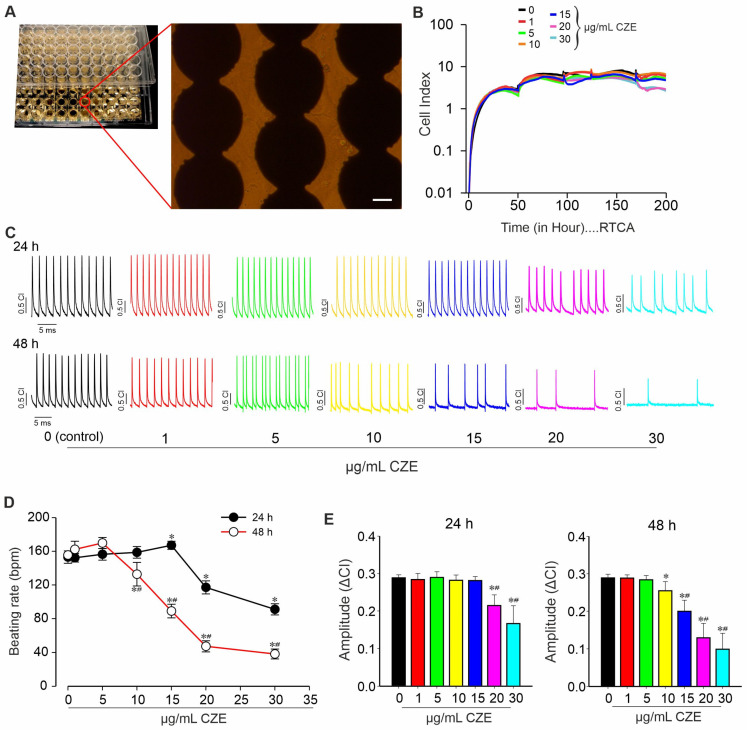
Effect of *Crinum zeylanicum* extracts (CZE) on cell index (CI) and spontaneous contractile activity of miPSC-CMs using xCELLigence RTCA Cardio System. (**A**) Representative images of a part of E-Plate Cardio 96 and zoomed in of a well containing monolayer of spontaneously beating iPSC-CMs growing on microelectrodes. (**B**) Representative graph from one of three independent experiments showing the effects of CZE extract on miPSC-CMs. The CI was monitored in real time for about 200 h after addition of 0 (control), 1, 5, 10, 15, 20, and 30 µg/mL CZE. (**C**) Representative beating rate pattern of 20 ms recordings of miPS-CMs 24 h (upper panel) and 48 h (lower panel) with various concentrations of CZE. (**D**,**E**) Average change in beating rate (**D**) and in the amplitude (delta CI) (**E**) of beating signals of iPSC-CMs under different concentrations of CZE. Asterisks indicate results that show significant difference from control by Student’s *t*-test. Scale bar = 50 μm. Results are reported as the mean ± SEM (*n* = 3 independent experiments). * and **^#^** (*p* < 0.05, Student’s *t*-test) indicate results that show significant difference from control (untreated) and from previous concentrations, respectively.

**Figure 5 pharmaceuticals-14-01208-f005:**
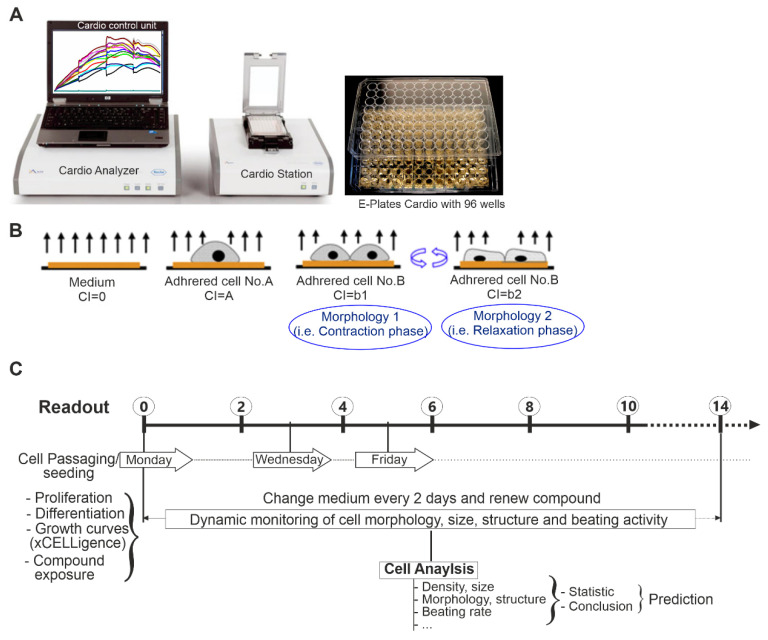
Real-time and dynamic assessment of cells viabibity, proliferation, differentiation, and function using xCELLigence system. (**A**) xCELLigence system technology: RTCA Cardio Station used in combination with the Cardio Analyzer and Control Unit. E-Plates Cardio with 96 wells, up to 80% of the bottom area of each well is covered with interdigitated gold microelectrodes. (**B**) The principle of real-time impedance monitoring (see methods section for more details). (**C**) Scheme showing the time course of the experiment.

## Data Availability

Data is contained within the article.

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
