# Peer review of "Assessment of the In Vitro Cytotoxicity Effects of the Leaf Methanol Extract of Crinum zeylanicum on Mouse Induced Pluripotent Stem Cells and Their Cardiomyocytes Derivatives"

_pharmaceuticals, 2021, doi:10.3390/ph14121208_

Round 1

Reviewer 1 Report

Dear authors,

For starters I want to congratulate you for the article.

General considerations:

  • The aim of the paper is clearly defined and concisely presented.
  • Regarding the methodology used, I appreciated first of all the very clear structure and the multitude of tests used, the presentation of the various stages of the experiment, significant figures.
  • The results are presented in a clear manner, with reference to literature data, respectively significant charts and images.
  • Chapter of discussion is drawn up in a clear and logical manner, making reference to the relatively recent literature data.

In addition to the positive aspects, there are also some negative aspects, which I believe need to be clarified and corrected in order to publish this manuscript:  

  • correct certain typing errors, unit of measurement, see lines 14,16, 18, 115. Font size, lines 173 - 174.
  • in the results chapter, line 82, I consider that the concentration of CZE extract must be written. In Figure 2A, the representation of the microplate, what does it mean on line H medium?
  • for figure 3A the image clarity / quality is not very good.
  • line 201 appears (ref ???). What does this mean?
  • I consider that it is important to appear some data related to the characterization of the extract used in the tests (concentration in total polyphenols, flavonoids, alkaloids).

Yours faithfully,

Author Response

Reviewer 1#

  • The aim of the paper is clearly defined and concisely presented.
  • Regarding the methodology used, I appreciated first of all the very clear structure and the multitude of tests used, the presentation of the various stages of the experiment, significant figures.
  • The results are presented in a clear manner, with reference to literature data, respectively significant charts and images.
  • Chapter of discussion is drawn up in a clear and logical manner, making reference to the relatively recent literature data.

Response:

We appreciate the reviewer for this opinion, and the useful observations that ensue.

In addition to the positive aspects, there are also some negative aspects, which I believe need to be clarified and corrected in order to publish this manuscript:  

  • correct certain typing errors, unit of measurement, see lines 14,16, 18, 115. Font size, lines 173 - 174.

Response: Corrected

  • in the results chapter, line 82, I consider that the concentration of CZE extract must be written. In Figure 2A, the representation of the microplate, what does it mean on line H medium?

Response: Corrected

  • for figure 3A the image clarity / quality is not very good.

Response: Corrected

  • line 201 appears (ref ???). What does this mean?

Response: Corrected

  • I consider that it is important to appear some data related to the characterization of the extract used in the tests (concentration in total polyphenols, flavonoids, alkaloids).

Response:

We thank the reviewer for raising this issue. For reviewer information, this is an extensive collaborative project of different teams who, since few years are working on some important and selected medicinal plants. The project aim to evaluate the antioxidant and cytotoxic activities of methanolic and aqueous extracts of the leaves/roots of this species, besides to obtain the chromatographic profiles and phytochemical screening of these plant extracts. In this project, our contribution is to determine the bioactivity of the extract and each compound isolated individually as well as in combination manner for each medicinal plant, with the aims to determine the bioactive fraction/compound/composition. This in close collaboration with our chemistry partner. Recently we published the results of a part of the phytochemical analysis done on CZE [1].The results revealed that leaves of CZE contain a higher total polyphenol, and flavonoid. A complete phytochemical screening of particular plant is underway and will be published after completion.

Reviewer 2 Report

Herbal medicine deals with the production of herbal medicines from natural or processed raw materials obtained from medicinal plants and their use in the prevention and treatment of diseases. Phytotherapy is also associated with the search for new plant medicines and the discovery of new phytotherapeutic applications of already known medicinal plants. Herbal medicine is very widespread among non-industrialized societies. It is the main component of all traditional medical systems (folk medicine), mainly due to the low cost of obtaining medicinal raw materials and their natural occurrence.
Crinum zeylanicum leaves are widely used in African folk medicine to treat cardiovascular ailments. The manuscript introduction is well written. The authors investigated the cytotoxic effect of methanol extract from Crinum zeylanicum leaves using pluripotent mouse stem cells. The expression of markers associated with the main types of heart cells was tested by immunofluorescence and quantitative RT-PCR.
The research was properly planned and described. The results showed that the plant extract significantly reduced cell proliferation and viability in a concentration and time dependent manner. High concentrations of Crinum zeylanicum have reduced the expression of some important cardiac biomarkers. At the same time, Crinum zeylanicum can be toxic at high concentrations in the embryonic stem cell stages and can modulate contractile activity.
The results presented in the manuscript are of great practical importance.
However, before publishing the manuscript, authors should complete the information on the chemical reagents used, i.e. their degree of purity.

Author Response

Reviewer 2#

Herbal medicine deals with the production of herbal medicines from natural or processed raw materials obtained from medicinal plants and their use in the prevention and treatment of diseases. Phytotherapy is also associated with the search for new plant medicines and the discovery of new phytotherapeutic applications of already known medicinal plants. Herbal medicine is very widespread among non-industrialized societies. It is the main component of all traditional medical systems (folk medicine), mainly due to the low cost of obtaining medicinal raw materials and their natural occurrence.
Crinum zeylanicum leaves are widely used in African folk medicine to treat cardiovascular ailments. The manuscript introduction is well written. The authors investigated the cytotoxic effect of methanol extract from Crinum zeylanicum leaves using pluripotent mouse stem cells. The expression of markers associated with the main types of heart cells was tested by immunofluorescence and quantitative RT-PCR.
The research was properly planned and described. The results showed that the plant extract significantly reduced cell proliferation and viability in a concentration and time dependent manner. High concentrations of Crinum zeylanicum have reduced the expression of some important cardiac biomarkers. At the same time, Crinum zeylanicum can be toxic at high concentrations in the embryonic stem cell stages and can modulate contractile activity.
The results presented in the manuscript are of great practical importance.
However, before publishing the manuscript, authors should complete the information on the chemical reagents used, i.e. their degree of purity.

Response:

We thank the reviewer for raising this issue. For this study, we ensured that CZE was totally free from solvent traces used during the extraction process. Nevertheless, phytochemical screening which involves purification, and characterization in addition to the botanical identification, extraction with suitable solvents (which does not causes degradation of the compounds) of the active constituents still need to be performed in order to further validate the bioactive compound of CZE.

Reviewer 3 Report

The authors performed an interesting study showing the citotoxic effects of the leaf methanolic extract of Crinum zeylanicum in cardiac-differentiated cells. Despite being in general well written, some parts need corrections before the acceptance by the journal (please check the PDF with my comments, corrections and suggestions). Some few parts need to be clarified and others corrected, the numbers of the figures should be changed and the discussion must be improved. A big part of it is composed by the repetition of the aim and results. I have only one major question: Could the authors perform a chromatography analysis to show the phytochemical composition of their extract? Or even some spectrophotometric analysis? It would enrich the paper if they could describe what there is inside the extract that could explain the biological activity. They discussed based on other studies, but some of these neither analyzed the same type of extract. In any case, I recommend minor review to the paper. 

Author Response

Reviewer 3#

The authors performed an interesting study showing the citotoxic effects of the leaf methanolic extract of Crinum zeylanicum in cardiac-differentiated cells. Despite being in general well written, some parts need corrections before the acceptance by the journal (please check the PDF with my comments, corrections and suggestions).

Response:

We thank the reviewer for his constructive and insightful corrections and comments, which have helped us to substantially improve our manuscript.

Some few parts need to be clarified and others corrected, the numbers of the figures should be changed and the discussion must be improved. A big part of it is composed by the repetition of the aim and results. I have only one major question: Could the authors perform a chromatography analysis to show the phytochemical composition of their extract? Or even some spectrophotometric analysis? It would enrich the paper if they could describe what there is inside the extract that could explain the biological activity. They discussed based on other studies, but some of these neither analyzed the same type of extract. In any case, I recommend minor review to the paper. 

Response:

We thank the reviewer for this important observation and suggestion. We have corrected the manuscript accordingly. As mentioned above, the all project aim to evaluate the antioxidant and cytotoxic activities of methanolic and aqueous extracts of the leaves/roots of CZE, besides to obtain the chromatographic profiles and phytochemical screening of this plant extract. In this project, our contribution is to determine the bioactivity of the extract and each compound isolated individually as well as in combination manner for each medicinal plant, with the aims to determine the bioactive fraction/compound/composition. This in close collaboration with our chemistry partner. Recently we published the results of a part of the phytochemical analysis done on CZE [1].The results revealed that leaves of CZE contain a higher total polyphenol, and flavonoid. A complete phytochemical screening of particular plant is underway and will be published after completion.
